# Physiological and Pathological Remodeling of Cerebral Microvessels

**DOI:** 10.3390/ijms232012683

**Published:** 2022-10-21

**Authors:** Pavel P. Tregub, Anton S. Averchuk, Tatyana I. Baranich, Maria V. Ryazanova, Alla B. Salmina

**Affiliations:** Federal State Budgetary Scientific Institution Research Center of Neurology, 125367 Moscow, Russia

**Keywords:** brain, plasticity, microcirculation, angiogenesis, microvessel regression, Alzheimer’s disease

## Abstract

There is growing evidence that the remodeling of cerebral microvessels plays an important role in plastic changes in the brain associated with development, experience, learning, and memory consolidation. At the same time, abnormal neoangiogenesis, and deregulated regulation of microvascular regression, or pruning, could contribute to the pathogenesis of neurodevelopmental diseases, stroke, and neurodegeneration. Aberrant remodeling of microvesselsis associated with blood–brain barrier breakdown, development of neuroinflammation, inadequate microcirculation in active brain regions, and leads to the dysfunction of the neurovascular unit and progressive neurological deficits. In this review, we summarize current data on the mechanisms of blood vessel regression and pruning in brain plasticity and in Alzheimer’s-type neurodegeneration. We discuss some novel approaches to modulating cerebral remodeling and preventing degeneration-coupled aberrant microvascular activity in chronic neurodegeneration.

## 1. Introduction

The last decade has been marked by agrowing interest in understanding the morphology and functional activity of brain microvessels, and their development in the prenatal period and throughout life [1]. There is growing evidence that the remodeling of cerebral microvessels plays an important role in plastic changes in the brain associated with its development, experience, learning, and memory consolidation. Neoangiogenesis is the process of the formation of new vessels from existing vascular structures. In the organism, it accompanies physiological (reparative processes and development) and pathological (malignant tumors, atherosclerosis, arthritis, hyperproliferative conditions) events. The brain microvascular tree also changes during ontogenesis, and cerebral neoangiogenesis is not a prerogative of the developing brain. Even in the adult brain, it is associated with the hippocampal activation in learning, the action of environmental enrichment or physical activity, and the migration of newly-born neurons from neurogenic niches to the loci of brain lesions [2].

Abnormal neoangiogenesis or deregulated regulation of microvascular regression could contribute to the pathogenesis of neurodevelopmental diseases, stroke, and neurodegeneration. Aberrant remodeling of microvesselsis associated with blood–brain barrier (BBB) breakdown, development of neuroinflammation, inadequate microcirculation in active brain regions, and leads to the dysfunction of the neurovascular unit and progressive neurological deficits [3].

There is a paradox that even though there is a huge number of studies aimed at describing cerebral (neo)angiogenesis, there is still a gap in the understanding of the regression of brain microvessels. However, the deregulation of microvascular remodeling in the brain tissue might be a significant contributor to the pathogenesis of Alzheimer’s disease [4], vascular dementia [5], and secondary parkinsonism [6]. Activation of neoangiogenesis and reduction in cerebral microvascular involution promotes angiogenesis in malignant neoplasms [7]. Since inhibition of microvascular involution and modulation of signaling pathways of aberrant angiogenesis has become a promising therapeutic strategy, understanding the mechanisms of pathological regression of microvessels is an extremely relevant topic. In addition, understanding the mechanisms of blood microvessels development and involution is required for the development of up-to-date in vitro tissue models of BBB and brain tissue. Particularly, cerebral organoids as promising brain in vitro models lack microvessels and are in extreme need of “artificial” vascularization for their proper activity.

In this review, we summarize current data on the mechanisms of blood vessel regression in brain plasticity and, particularly, in Alzheimer-type neurodegeneration. We should note here that there are some studies related to the regression of lymphatic microvessels that might have some similarities with blood microvessels [8,9], including those in the central nervous system [10]. Thus, some data in the current review will rely on common signaling molecular pathways in the life cycle of cells present in these two types of microvessels. Finally, we discuss some novel approaches to modulating cerebral microvascular remodeling and preventing the degeneration-coupled aberrant activity of brain capillaries in chronic neurodegeneration.

## 2. General Mechanisms of Microvasculature Remodeling

The regression of blood vessels in various tissues is a multifactorial complex process that is still less understood compared with angiogenesis. The regression is usually explained in the context of trimming microvessels during their maturation. Vessel regression includes a diverse set of molecular markers and pathways [3,11]. Korn and Augustin [3] described several mechanisms of vascular regression, with a special focus on the role of vascular endothelial growth factor (VEGF), Wnt/Notch-pathway, angiopoietin (Ang)/tyrosine kinase with immunoglobulin-like and EGF-like domains (TIE)-signaling, Notch/delta-like canonical Notch ligand 4 (Dll4), and other regulatory factors. They suggested that the regression of vessel branches within the microvascular bed is controlled by blood flow, and a blood vessel should be occluded until the blood flow is completely stopped. Endothelial cells within regressing vascular segments may retract and undergo apoptosis or migrate to other locations, thereby leaving behind empty basement membrane areas.

It is well known that in humans, the VEGF family consists of several members: VEGF-A (different isoforms), VEGF-B, VEGF-C, VEGF-D, VEGF-E/viral VEGF, VEGF-F (a factor from the venom of some snakes), PlGF (placental growth factor), and EG-VEGF (endothelial growth factor of endocrine glands) [12]. VEGF binds to protein tyrosine kinase receptors (VEGFR) of three types. VEGF-A protein binds to the VEGFR-1 (Flt-1) and VEGFR-2 (KDR/Flk-1) receptors. In addition, VEGFR-2 mediates almost all known cell responses to VEGF (angiogenesis, macrophage/granulocyte chemotaxis, and vasodilation). The functions of the VEGFR-1 receptor are not well-described yet. It is assumed that it modulates VEGFR-2 signaling. VEGF-C and VEGF-D proteins are the ligands for the VEGFR-3 receptor known as a signaling molecule in lymphatic angiogenesis.

According to Lobov et al. [13], modulation of VEGF-A levels and Dll4/Notch-signaling induces distinct changes in blood vessel morphology and gene expression, thereby indicating that these pathways may be largely independent. However, in the Wnt/Ca2+/Nfat signaling pathway, endothelial R-spondin-3 (RSPO3) activates Wnt signaling for further up-regulation of VEGF gene expression needed for angiogenesis [14,15,16]. It was shown that VEGF is an immediate early gene in angioblasts [17], and therefore, Wnt/VEGF signaling could be the key regulatory mechanism of angiogenesis-associated endothelial cell migration [18] and shear stress sensing [19]. Moreover, recent data reveal that in cerebral endothelial cells, the permeability of the cell layer is under the control of VEGF-driven changes in the expression of numerous genes [20]. Wnt signaling is required for the proliferation of stalk cells in the developing vessel, whereas Dll4 expression in activated stalk cells results in the up-regulation of Wnt signaling in adjacent tip cells [21]. Notch signaling in sprouting endothelial cells suppresses cell migration, supports barriergenesis [22], and keeps the balance of apoptosis and endothelial cell survival [23]. Dll4/Notch signaling, in general, limits the migration of tip cells attracted by high local levels of VEGF-A [24], but VEGF itself is capable of inducing the expression of Dll4 and Notch in microvascular endothelial cells [25]. In sum, the remodeling of microvessels is driven by the reciprocal effects of VEGF, Notch, and Wnt on their expression in tip and stalk cells. VEGF plays a key role in the stimulation of cell migration/proliferation followed by the suppression of angiogenic events and establishment of barrier integrity.

Taking into consideration the well-described role of VEGF in the hypoxic response, it is tempting to speculate that vascular remodeling might be simply controlled by tissue oxygen availability. If so, a low oxygen supply promotes (neo)angiogenesis, while a high oxygen supply stimulates vessel involution. Indeed, it is well-known that hypoxia-inducible factor 1-alpha (HIF-1α) accumulates in cells under hypoxic conditions, which leads to the transcription of pro-angiogenic genes, including VEGF-A and VEGF receptors [26]. Thus, it is reasonable that an insufficient rise in local VEGF levels leads to the regression of the microvascular bed. However, the mechanism might not be so simple since other factors are no less important. For instance, the accumulation of protein C in the brain tissue in the postischemic period stimulates endothelial proliferation, increases vascular density, and activates pro-angiogenic integrins α5β1 and avβ3 [27]. In parallel, protein C suppresses apoptosis of brain endothelial cells in ischemia [28], thereby preventing the reduction of the capillary bed.

Moreover, the establishment of new vessels or their disappearance could be a derivative of a tissue response to hyperoxia. The murine oxygen-induced retinopathy model is a tool that allows studying the regression of blood vessels under conditions of hyperoxia. Claxton and M. Fruttiger [29] used a model of induced hyperoxia in mice, in which VEGF induction was suppressed, to demonstrate that fibroblast growth factor 2 (FGF2), angiopoietin-2 (ANG2), platelet-derived growth factor (PDGF), and DLL4 are independently involved in maintaining the stability of vascular growth and development. Zhong et al. [30] developed an animal model for studying the process of growth and regression of blood vessels using transgenic mice-Prox1-GFP/Flt1-DsRed (PGFD). This model allowed direct visualizing of the development, branching, and regression of both vessel types (lymphatic and blood) in various organs by means of confocal and two-photon microscopy. They found that deletion of VEGFR2 abolished VEGFA- and VEGF-C-induced corneal lymphangiogenesis. Thus, elevated local levels of VEGF may promote (neo)angiogenesis, whereas lowered levels of VEGF support the regression of blood and lymphatic microvessels.

Expression of the VEGF gene is mainly under the control of HIF-1-driven transcription. Since HIF-1-signaling controls inflammation caused by inadequate tissue oxygen supply, it is not surprising that there is a close relationship between inflammation and angiogenesis. At least two proinflammatory molecules, interleukin-8 (IL-8) and cyclooxygenase-2 (COX-2), affect angiogenesis in inflamed tissue [31]. Microvessel remodeling is modulated by anti-inflammatory agents: glucocorticoids as suggested by Logie et al. [32], or COX-2 inhibitors. There are data confirming that capillary regression is under the control of tissue inhibitors of metalloproteinases TIMPs-2, 3, and 4 [33], which are well-known contributors to the inflammation-driven remodeling of the microvascular basement membrane and extracellular matrix.

A concept called anti-angiogenic switch leading to vessel regression has been proposed as a negative feedback mechanism protecting endothelial cells from excessive angiogenesis and overstimulation by VEGF [34]. However, there is a gap in understanding the entire mechanism of feedback control. On one hand, it might be mediated by Dll4/Notch signaling suppressing cell migration, as discussed above. If the switch is initiated, even high concentrations of exogenous VEGF, FGF, and PDGF do not prevent the regression of blood vessels [35], whereas proteins of the Vasohibin and Sprouty families provide anti-angiogenic effects [36,37,38]. Presumably, the basic mechanism of the involution of the microvascular bed is endothelial apoptosis [39,40]. However, Franco et al. [41] studied blood vessels in the cornea of mice and zebrafish under hyperoxic exposure and concluded that apoptosis does not contribute significantly to vascular regression. In addition, they reported that the migration of endothelial cells into neighboring vessels might be important for vascular regression. It is known that the regression of hyaloid vessels and pupillary membranes under physiological conditions is coupled to the apoptosis progression and modulation of the cell cycle [42,43,44,45]. In the context of apoptosis-mediated signaling in vascular regression, activation of CXC chemokine receptor 3 (CXCR3) is an important component [46]. Data obtained by the group of Richard J. Bodnar [47] showed that newly formed vessels express CXCR3 receptors that trigger the μ-calpain-mediated cleavage of the cytoplasmic tail of β3 integrins. Activation of CXCR3 by the ligand IP-10 (CXCL10) inhibits the development of a new vasculature and causes regression of newly formed vessels [48]. Calpain-dependent cleavage of β3 integrin results in the suppression of cell adhesion and promotes apoptosis of endothelial cells [49], whereas activation of β3 integrins is required for angiogenesis [27]. Thus, apoptosis of endothelial cells, loss of their contacts with the extracellular matrix, suppression of tipcell migration, and proliferation of stalkcells caused by the decline in VEGF activity and deregulation of Dll4/Notch signaling may result in the involution of microvessels.

The presence of the plexin–semaphorin–integrin domain in semaphorins that have been implicated in the suppression of integrins activity and tube formation in endothelial cells, suggests their involvement in the down-regulation of neoangiogenesis [50]. Indeed, studying the role of semaphorins (Sema) expressed by endothelial cells in the remodeling and regression of microvessels has received special attention [49,50,51,52,53,54]. Chen et al. [54] demonstrated that HIF-2α directly regulates Sema3G transcription in endothelial cells during hypoxia. Sema3G, in turn, coordinates the functional interactions between cell adhesion molecules, β-catenin and VE-cadherin in the endothelium. Administration of semaphorin 3G (Sema3G) stimulates the regression of dysfunctional vessels, thereby promoting the formation of a healthy vasculature. Deletion of stress-induced transcription factor NF-E2-related factor-2 (Nrf2) leads to the induction of semaphorin 6A (Sema6A) expression in hypoxic/ischemic ganglion cells via HIF-1α-mediated mechanism and suppression of endothelial cells migration through Notch signaling. By contrast, activation of Nrf2 promotes reparative angiogenesis and reduces pathological neovascularization [49]. In cerebral arteriovenous malformations, the interplay of HIF-1/VEGF and Nrf2 signaling cascades contributes to angiogenesis: loss of Nrf2 inhibits the effects of VEGF and results in a lower migratory activity of endothelial cells [55]. In rats with the models of traumatic brain injury, Nrf2 activation is required for the preservation of the structural integrity of the BBB [56]. Thus, the common triggers of microvascular remodeling in various tissues are hypoxia/ischemia or hyperoxia, inflammation, injury, and other stress factors that cause apoptosis/detachment of endothelial cells and enhance the permeability of the endothelial barrier. All these mechanisms are aimed at establishing a functionally competent microvascular network (angiogenesis), and eliminating microvessels with aberrant integrity of the endothelial cell layer (microvascular involution).

## 3. Functional Competence of Cerebral Endothelial Cells and Their Role in Brain Vessel Remodeling

In the postnatal brain, the development of microvessels occurs mainly due to the formation of new vessels from the pre-existing vasculature without the significant contribution of endothelial progenitor cells [57]. Endothelial cells lining the vascular network in the brain differ from peripheral endothelial cells. Particularly, brain microvessel endothelial cells (BMECs) are: (i) surrounded by pericytes and astrocytes that tightly control their paracellular permeability and adjust it to the activity of neurons within the neurovascular unit; (ii) coupled via numerous tight and adherence contacts that limit paracellular permeability within the blood–brain barrier; (iii) known to have fewer fenestrae than peripheral endothelial cells and higher diversity of transmembrane transporters that control the transcellular permeability of the barrier; (iv) enriched with mitochondria and demonstrate extensive oxidative metabolism matching their functional activity [58]. Also, BMECs are differently subjected to shear stress compared to peripheral vessels [59], which is needed for their functional activity, metabolic plasticity, adhesion, and response to the action of numerous regulatory molecules [60]. It has been shown that shear stress-regulated inward rectifier potassium channels (Kir2.1) in retinal endothelial cells are indispensable in microvessel maturation and pruning of excess vessels [61]. Moreover, shear stress controls the polarity and alignment of endothelial cells (HUVEC) by acting at the mechanosensitive complex VEGFR–CD31–VE-cadherin [62], but whether this mechanism is active in BMECs remains to be assessed.

In embryos, BMECs originate from progenitor cells of the primary canal of the midbrain. They form a vasculature due to a combination of targeted migration and controlled sprouting [63,64]. K.I. Boström and colleagues postulated that the tissue-specific vascular endothelial cells (including BMECs) may have tissue-specific origin [65]. Using flow cytometry, the authors found that a group of endothelial cells at the early stages of embryonic brain development co-expresses SOX2 (a key regulator of neuronal differentiation and brain development) and the endothelial marker VE-cadherin. Therefore, the common origin of these two cell types may allow understanding of the development of cells within the neurovascular unit [58]. Experimental data obtained on cerebral organoids established from the induced pluripotent stem cells (iPSCs) that lack the ability for vascularization argue against this proposal [66]. However, the development of the microvascular bed and neuronal network in the embryonic brain seem to have many shared properties and molecular regulators [67,68]. Recently, new data have been obtained on the contribution of neurons to the regulation of local blood flow in active brain regions: stimulation of transient receptor potential cation channels (TRPA1) in BMECs by neuronal activity results in the dilation of ascending arterioles [69]. Presumably, these channels could serve as sensors of neuronal activity that initiate vasodilation to redirect blood flow to the regions of enhanced oxygen needs. There are data that TRPA1 is a critical factor for angiogenesis in tumor-derived endothelial cells [70]. If a similar mechanism exists in BMECs, it might link the neuronal activity with the (neo)angiogenesis in the brain.

Being stimulated by global or local hypoxia, BMECs initiate angiogenesis. As we have mentioned before, the role of endothelial progenitor cells in brain neoangiogenesis is still under debate [71]. Thus, BMECs are capable of completing the angiogenesis program. For instance, mature endothelial tip cells undergo a metabolic switch to enhance glycolysis and migrate along the gradient of pro-angiogenic molecules. Stalkcells accelerate oxidative phosphorylation in mitochondria, proliferate extensively, and establish newly-formed microvessels [72,73]. The expression profile of BMECs during angiogenesis is characterized by a certain set of active genes [74], mainly linked to the extensive Wnt/β-catenin signaling and oxidative metabolism [75]. The transcriptional profile of activated BMECs demonstrates significant changes similar to their response to oxidative stress or the action of inflammatory stimuli [76].

An important feature of endothelial cells that affects angiogenesis is their quickly adjustable metabolism [77]. According to the current concept, glycolysis, which is the main mechanism of energy supply in endothelial cells [78], is controlled by 6-phosphofructo-2-kinase/fructose-2,6-bisphosphatase-3. It generates energy to maintain the competitive behavior of endothelial cells at the tip of the growing vascular process. Fatty acid oxidation is controlled by carnitine palmitoyltransferase 1a. It regulates nucleotide synthesis and endothelial stalkcell proliferation [77]. Recent data reveal [79] that the hCMEC/D3 endothelial cell line (originated from human brain microvessels) demonstrates the prevalence of glycolytic ATP generation even in the presence of abundant oxygen.

Another important distinguishing feature of BMECs is the coupling of angiogenesis and barriergenesis [80]. The restrictive architecture of the BBB reduces paracellular diffusion while the activity of controlled transporters and endocytosis in BMECs limits transcellular transport. Thus, the establishment of new microvessels in the brain tissue requires concomitant maturation of BMECs with the fully expressed set of transporters and junction proteins. One of the most studied regulators of tight junction machinery is protein kinase C. Inhibition of protein kinase C activity using its dominant-negative mutants reduces the effect of monocyte chemoattractant protein-1 (MCP-1 or CCL2) on the permeability of the brain endothelium [81]. Other mediators of intercellular signaling in BMECs are protein kinase A, protein kinase G, Ras homolog gene family member A, mitogen-activated protein kinase, JNK-kinase, phosphatidylinositol 3-kinase/Akt, and Wnt/β-catenin pathways [82,83]. Thus, maturation and establishment of the barrier function in the BBB are driven by the phosphorylation of junction proteins, the establishment of the perivascular cellular microenvironment (pericytes and astrocytes), and the acquisition of the corresponding metabolic status of BMECs.

We have shown before that maturity of BMECs is partially under the control of HIF-1 expression in perivascular astrocytes [84]. The molecular mechanism of barriergenesis, which has been reviewed elsewhere [85], includes the activity of numerous growth factors, neuromediators, and cytokines released from the cells of the neurovascular unit: VEGF, PDGF, Ang-1, cyclophilin, MMP, etc. As a result of their action, neoangiogenesis is initiated at sites of higher metabolic needs (e.g., in constitutively active brain regions). Concomitant microvessel pruning allows the elimination of the vessels with immature BBB (to prevent neuroinflammation) or insufficient blood perfusion (to prevent ischemia). Indeed, pruned segments of brain microvessels exhibited a low and variable blood flow, which further decreased irreversibly prior to the onset of pruning [86]. When brain neoangiogenesis is too fast and extensive, for instance, in post-stroke conditions or the initial phase of neurodegeneration, microvessels might acquire non-completed barriergenesis. In BMECs, this is associated with aberrant shear stress response, increased apoptosis, and elevated detachment from the basement membrane. As a result, there is an enhanced BBB permeability and a promotion of neuroinflammation [87]. Differentiation and maturation of cells always require an action of signaling molecules promoting cell survival, whereas the elimination of immature or damaged cells leads to the induction of apoptosis. Therefore, the involution of microvessels could be predominantly controlled by BMEC apoptosis.

## 4. Brief Overview of BMEC Apoptosis and Microvessel Pruning

Apoptosis of BMECs usually occurs in capillaries with critically lowered perfusion. There is a phenomenon of so-called “string vessels” [88] that are empty basement membrane strands without BMECs and perfusion that are formed after the death of endothelial cells in the brain tissue. Such vessels have defective coverage with astrocytes and pericytes, and thus they cannot respond to the activity of neurons within the neurovascular unit [89]. String brain microvessels, probably, may act as tunneling nanotubes involved in vascular remodeling and reparative angiogenesis [90]. String vessels are found in the central nervous system more often than in other organs, and gradually disappear over the time-course of months or years [91]. They are associated with the progression of Alzheimer’s disease, ischemia, and radiation-induced brain injury, but may also be found in the normal human brain at any stage of its development. It is likely that localization and the number of string vessels in the brain tissue could predict the development of chronic neurodegeneration.

Apoptosis and relocation of BMECs play a role in the mechanism of physiological regression of microvessels during remodeling [3]. In the larval zebrafish, apoptosis of BMECs contributes to vessel pruning during development: in contrast to brain vessels undergoing BMEC migration-associated pruning, apoptosis-mediated pruning results in the appearance of much longer and highly permeable microvessels. It has been suggested that long vessels limit the ability of BMECs to migrate into adjacent unpruned segments, therefore BMECs undergo apoptosis at their initial location [92]. As a result, string vessels might appear.

It is believed that the physiological turnover of brain microvessels is very low in the postnatal/adult brain, but it is accelerated in aging [93]. However, in the adult or aging brain, the death of BMECs under pathological conditions occurs via apoptosis or, less frequently, via autophagy, lysosomal degradation, or necroptosis [94]. Several factors can induce BMEC apoptosis in pathology: prooxidants, pro-inflammatory cytokines, chronically elevated glucose, bacteria and viruses, toxic xenobiotics, hormones, ionizing radiation, aberrantly folded proteins, etc. [95,96,97]. For instance, Anasooya et al. [98] demonstrated that different concentrations of hydrogen peroxide play an important physiological role, but under pathological conditions, higher levels of hydrogen peroxide induce massive apoptosis of BMECs. Being applied at concentrations of less than 1 µM, hydrogen peroxide increases tube formation, but acting at concentrations of 10 mM and above, it reduces cell viability and induces apoptosis of BMECs in vitro. The findings of Shao and Bayraktutan [99] revealed that hyperglycemia promotes apoptosis of BMECs by means of induction of protein kinase Cß_I_ (PKC) and the reactive oxygen species-generating enzyme NADPH oxidase. Apoptosis of BMECs can be significantly affected by the activation of enolase-phosphatase 1 (ENOPH1) which increases the formation of reactive oxygen species, activation of pro-apoptotic proteins, and DNA repair-modulating proteins (caspase-3, PARP, BAX) as well as endoplasmic reticulum stress proteins (IRE1, GRP78, PERK, and calnexin) in endothelial cells upon glucose–oxygen deprivation [100]. Inhibition of PKC prevents BBB breakdown in the conditions of oxygen–glucose deprivation due to the suppression of cytoskeletal rearrangements and apoptosis in BMECs [101].

Activation of caspase cascade in BMECs depends on multiprotein complexes that include various factors, such as apoptotic protease activating factor 1 (APAF1) and cytochrome C (CytC) released into the cytosol (Figure 1). One of the key proteins that transmit apoptotic signals is Fas-associated protein with death domain (FADD) which is activated by: (i) binding to FAS/procaspase-8 and formation of a death-inducing signaling complex (DISC) with subsequent activation of caspases 3, 6, and 7; (ii) recruitment of procaspase-10 and caspase-8/-10 regulator cFLIP [102]. After induction of apoptosis, caspase-3 cleaves the DFF45/ICAD complex and releases DFF40/CAD. It leads to DNA fragmentation and nuclear condensation [103]. An elevated ratio of the proapoptotic protein BAX and antiapoptotic protein BCL-2 levels [104], nuclear translocation of apoptosis-inducing factor (AIF), overactivation of poly(ADP-ribose) polymerase-1 (PARP-1) leading to NAD+ depletion and metabolic stress also contribute to BMEC apoptosis [105]. BMECs might be particularly sensitive to NAD+ depletion because of their high energetic demands [71]. That is why replenishment of intracellular NAD+ levels is beneficial in preventing cerebral vascular aging [105], or in protecting BMECs from hydrogen peroxide-induced apoptosis in vitro [106].

Microglia and macrophages induce the death of BMECs via the release of apoptosis- or necroptosis-inducing factors. This phenomenon is implicated in vascular remodeling [107,108], or recognition of apoptotic bodies derived from damaged BMECs [92]. Recent experimental data reveal that capillary-associated microglia engulf apoptotic bodies and greatly affect brain microvasculature: elimination of this type of microglia results in a ~15% increase in capillary size, a ~20% increase in cerebrovascular perfusion, and a ~50% reduction in vascular reactivity [109]. It is interesting that like in the case of microglia-mediated synaptic pruning mediated by C3 complement and its receptor C3R [110], vessel pruning performed by microglia depends on the activity of the C3/C3R-system [111]. Thus, vascular density in the brain is tightly coupled not only to the neuronal activity in the particular brain regions, but also corresponds to the activity of capillary-associated microglia and the intensity of the local immune reactions.

It should be emphasized that modulation of endothelial cell apoptosis is of great therapeutic importance. Therefore, there have been numerous attempts to utilize this phenomenon for the development of reparative strategies in brain pathology. One example is the use of FTY720, a sphingosine-1-phosphate receptor 1 (S1PR1) modulator [112], which restores the structure of the neurovascular unit after experimental traumatic brain injury by reducing BMEC apoptosis and attenuating the activation of glial cells. Another promising approach for inhibiting BMEC apoptosis is interleukin-10, whose anti-apoptotic effects are associated with the suppression of BAX and caspase-1 and 3, as well as with an increase in BCL-2 levels in endothelial cells in neuroinflammation [113].

## 5. Regulation and Outcomes of Vascular Regression and Pruning in the Brain

The process of regression of the microvascular network has been studied in detail in corneal vessels [114,115] as well as in embryonic tissue, endometrium, and in the models of wound healing [3,116,117,118]. It is commonly accepted that vessel regression is determined by genetic and epigenetic factors, pro- and anti-angiogenic environment, oxygen supply, and local blood flow. However, there are limited data demonstrating how brain blood vessel regression is changed with age, and how it affects neuronal activity [119,120,121].

Presumably, blockage and recanalization of capillaries in the cerebral cortex is a continuous process that could be changed with age through a VEGF-mediated mechanism [122]. As we mentioned above, regression of cerebral microvessels occurs both due to the migration of endothelial cells and activation of apoptosis [3]. According to Hughes and Chang-Ling [123], the physiological regression of brain blood vessels is predominantly mediated by the migration of BMECs further contributing to new vessels. When regression leads to impaired blood circulation but the relocation of BMECs is impossible, they undergo apoptosis.

There is a Wnt/β-catenin signaling pathway specific to angiogenesis in the CNS [60,124]. Defects in this pathway reduce the number of vessels, lead to the loss of tiny capillaries, and the formation of hemorrhagic vascular malformations that remain adherent to the meninges. The recent review by Gupta et al. [60] emphasizes that in the mechanism of cerebral microvessel remodeling, canonical signaling along the Wnt/β-catenin pathway in BMECs functions in a cell-autonomous manner and promotes the formation and maintenance of the BBB through specific protein–protein interactions. In addition, signaling via the Wnt/β-catenin pathway regulates the expression of the BBB-specific glucose transporter GLUT1 which is needed for better adjustment of BMECs to the elevated needs in ATP for vascular remodeling and barriergenesis [124].

The data obtained by Gao et al. [125] on in vivo longitudinal imaging showed that regressing vessels in the brain are widespread in mice, monkeys, and humans. The regression of these vessels proceeds through successive stages of blood flow occlusion, the collapse of endothelial cells, displacement or loss of pericytes, and retraction of glial ends. The authors found that short-term occlusion of the middle cerebral artery and lipopolysaccharide-mediated neuroinflammation induce an increase in vascular regression followed by metabolic impairments and reduction of neuronal activity.

It is well known that astrocytes play a key role in many aspects of vascular development and functioning. During brain development, the growth of blood vessels follows the establishment of the astroglial syncytium, which is formed due to the expression of intercellular channels—connexons. Deletion of the ORC3 gene in glial progenitors dramatically reduces the number of astrocytes in the early postnatal cerebral cortex, and this, in turn, leads to a serious decrease in both the density and branching frequency of cortical blood vessels [126]. However, this process is not accompanied by vascular regression, and the morphology of blood vessels in the mutant cortex is restored at later stages, after the occurrence of astrogliosis. Thus, these data indicate that astrocytes have pro-angiogenic properties without directly affecting the brain microvessel regression.

Astrocytes, pericytes, and capillary-associated microglia secrete a huge number of molecules with pro- or anti-angiogenic properties. In some cases, the pro- or anti-angiogenic potential of the same molecule changes in various conditions. For instance, locally released pro-angiogenic molecules (products of COX-2 activity, or angiopoietin-2) support angiogenesis [127]. However, experimental data suggest that angiopoietin-2 depending on the presence of VEGF can affect both angiogenesis and vascular regression in the brain. Moreover, the TWEAK protein (TNF-like weak inducer of apoptosis) which is elevated in the brain tissue upon restoration of normoxia, does the same [128]. Semaphorins (Sema6A, Sema3A, Sema3E, Sema3C, and Sema4G) expressed in the brain tissue also affect both neoangiogenesis and regression of macro- and microvasculature due to their effects on proliferation, apoptosis, and migration of BMECs and pericytes [129,130].

What is the key role of microvessel regression in the brain tissue? The establishment of new highly-branched microvessels seems to be usually associated with incomplete maturation of BMECs, and local BBB breakdown. Later, when brain metabolic requirements return to the pre-stimulated levels, the density of capillaries should also be reduced in order to prevent inadequate blood supply and risk of oxidative stress induction. Thus, the pruning of microvessels should be completed by means of BMEC migration or apoptosis [131,132]. Like cerebral angiogenesis in the postnatal brain, microvascular regression is associated with plasticity mechanisms. Indeed, the intensive formation of new microvessels under physiological conditions, for instance, during physical activity or extensive training, is induced by the action of VEGF and lactate. Lactate is glycolytically produced and released from astrocytes or pericytes, and likely acts at lactate receptors GPR81 widely expressed in BMECs [133].

Neoangiogenesis, in addition to neurogenesis, is critically important for effective learning and memory consolidation [134]. It is most probably determined by high metabolic demands in the active neurogenic niches and other brain regions [135]. Obviously, the cessation of such a request should lead to a gradual regression of the microvascular density. Indeed, elegant experimental data confirm that remote memory formation is associated, upon encoding, with a hypoxic signal that triggers angiogenesis in specific cortical regions. They support memory storage and further regression of recently-formed (or pre-existing) microvessels [136]. Moreover, permanent tuning of local blood flow in the brain is necessary either when adapting to oxygen or nutrient deficiency, and when neuronal activity is intensified in neuronal networks with spatial and temporal heterogeneity [137]. In addition to the rapid physiological responses of blood flow to reduced oxygen/substrate availability and increased brain energy demand, mechanisms of vascular plasticity should be of high importance. As we mentioned earlier, a key regulatory factor might belong to the HIF family of transcription factors. HIF-1α increases the production of VEGF and glycolytic flux resulting in the higher density of the microvascular network, whereas HIF-2 provides a compensatory response of cells to hypoxia. Both factors are involved in the regulation of brain development, synaptogenesis, neuritogenesis, and neurogenesis, being highly expressed in the embryonic and postnatal brain [138]. In endothelial cells, HIF-2 expression is necessary for the correct angiogenesis and cell adhesion controlling limited vascular permeability [139]. Like in the case of HIF-1, HIF-2 expression deficiency is also associated with cognitive impairments and aberrant metabolism in Alzheimer’s-type neurodegeneration [138,140].

## 6. Aberrant Microvessel Remodeling in Alzheimer’s-Type Neurodegeneration

Impaired microvascular remodeling, particularly aberrant microvessel regression, is recognized now as an important mechanism implicated in the pathogenesis of many brain disorders [141]. Reparative angiogenesis in the peri-stroke area is insufficient if it is not accompanied by reduced regression of formed vessels in the peri-infarction loci [142,143]. Chronic hypoxia, which induces vascular remodeling in the brain tissue, is accompanied by the accumulation of fibrinogen and activation of inflammatory response in non-angiogenic vessels [144]. This phenomenon could probably contribute to the progression of neuroinflammation, activation of microglia, and extensive microvascular pruning, as discussed above.

Neurovascular dysfunction is a hallmark of the neurodegenerative process associated with cerebral amyloid angiopathy caused by the accumulation of β-amyloid peptide in the brain tissue and walls of cerebral vessels [145,146]. Vascular changes occur even at the preclinical phase of Alzheimer’s disease before the development of cognitive impairments, and accumulation of beta-amyloid and hyperphosphorylated tau protein. They are always associated with the altered structural and functional integrity of the BBB [147]. In recent decades, special attention has been paid to the mechanisms of BBB dysfunction and breakdown, reduced cerebral blood flow, and impaired vascular clearance of beta-amyloid from the brain to meningeal lymphatic vessels [4,148,149,150]. Numerous data confirm that aberrant angiogenesis and aging of the cerebrovascular system can initiate neurovascular events leading to Alzheimer’s-type neurodegeneration [151,152,153].

Both, the density of cerebral vessels and the number of neurons decrease significantly during physiological aging (by 10–30%) and in patients with Alzheimer’s disease (by 40–60%) [5,154,155,156]. In turn, a decrease in the density of the microvascular bed serves as a prognostic marker in progressive Alzheimer’s-type neurodegeneration [157,158,159]. The pathology of the brain microvasculature in Alzheimer’s disease corresponds to the excessive deposition of beta-amyloid and tau protein [160,161,162]. However, a genetic predisposition to Alzheimer’s disease due to the abnormal APOE-ε4 genotype or family history is not associated with the presence of microvascular pathology in middle age [163]. At the same time, the data provided by Moore et al. [164] show an interplay of aberrant APOE-ε4 and VEGFR co-receptor neuropilin-1 (NRP1)/VEGF-A expression and impairment of cognitive functions: NRP1 reduces the risk of negative cognitive performance in carriers of the APOE-ε4 allele that might have a direct relation to its pro-angiogenic activity.

There is an altered balance of angiogenesis and vessel regression in Alzheimer’s disease. In the experimental models of Alzheimer’s disease, BMEC damage and apoptosis lead to deregulated angiogenesis [151]. BMECs in Alzheimer’s type neurodegeneration express extremely low levels of the mesenchymal homebox 2 (MEOX2) gene which regulates differentiation and remodeling of vascular cells and is restricted to the vascular system in the adult brain. Low levels of MEOX2 expression mediate abnormal angiogenic responses of BMECs to VEGF and other pro-angiogenic factors [165]. Reduced MEOX2 expression leads to accelerated vascular involution, inadequate blood supply, and BBB breakdown. In addition, low levels of MEOX-2 expression promote proteasomal degradation of the apolipoprotein E receptor (LRP1), further leading to a reduced ability to transport the excess beta-amyloid from the brain tissue to the peripheral blood [166]. Accumulation of beta-amyloid on the outer membrane of blood vessels, in turn, is an anti-angiogenic factor itself. It may contribute to a decreased density of the microvascular bed seen in Alzheimer’s disease [167,168]. Plasma concentrations of high molecular weight fibronectin, which binds integrins and has an angiogenic effect on BMECs, correlate with the risk of developing Alzheimer’s disease [169]. Aberrant remodeling of brain microvasculature results in defective clearance and excessive accumulation of beta-amyloid in the brain, cerebral blood flow decrease, and secondary metabolic alterations supporting neuronal dysfunction, cell death, and inflammation [170]. Excessive hypervascularization seen in Alzheimer’s disease is induced by the action of beta-amyloid on BMECs and leads to abnormally enhanced permeability of the BBB in newly formed vessels [87]. Thus, one may assume in the time course of Alzheimer’s neurodegeneration, the initial increase in cerebral angiogenesis later associates with accelerated vascular aging, altered microvascular regression, and preservation of immature microvessels with uncontrolled high BBB permeability. If so, the aberrant balance of angiogenesis and microvessel involution as well as impaired barriergenesis should be considered as promising targets for the prevention and compensation of neurological deficits in Alzheimer’s-type neurodegeneration.

## 7. Some Perspectives of Targeting Abnormal Microvascular Remodeling in Brain Diseases

Understanding key mechanisms of normal and aberrant cerebral microvessel remodeling enables further progress in the development of novel approaches to the diagnostics and treatment of brain disorders. For instance, elevated plasma levels of angiogenin and tissue inhibitors of matrix metalloproteinase-4 as well as VEGF have been identified as risk factors for the development of Alzheimer’s disease [153,171]. Application of exogenous activators of protein C, whose deficiency in the postischemic period leads to the inhibition of endothelial integrins α5β1 and avβ3, is considered to be a promising method of stroke therapy [27]. M. Ali and O. Bracko [172] recently found that upregulation of VEGF-A signaling contributes to a decrease in cerebral capillary blood flow in an experimental model of Alzheimer’s disease. Injection of anti-VEGF-A antibodies results in the improvement of BBB integrity, a decrease in the number of non-functioning capillaries, and restoration of cerebral blood flow. Durrant et al. [173] provided new data on beta-secretase as a potential therapeutic target for reducing aberrant angiogenesis in Alzheimer’s disease: in mice expressing human amyloid precursor protein gene, inhibition of beta-secretase normalizes excessive formation of endothelial filopodia and restores Notch signaling. Inhibition of excessive angiogenesis with the application of cytostatic drugs reduces the neurological manifestations of experimental Alzheimer’s disease [174]. Control of inflammation and insulin resistance achieved by the suppression of inflammasome formation is a way to prevent vascular cognitive impairments [175]. Other approaches to suppressing aberrant angiogenesis and microvessel regression in brain diseases are based on the application of pharmacological inhibitors of hypoxia-driven events, BMEC apoptosis, extracellular traps of neutrophils, canonical/non-canonical Wnt/β-catenin signaling pathways, and mechanisms contributing to vascular pathology in brain diseases [176,177,178,179] (Figure 2). However, it should always be taken into consideration that physiological signaling pathways might be completely deregulated in the brain pathology, thereby leading to unexpected outcomes of the proposed treatments.

In sum, cellular and molecular mechanisms of remodeling brain microvessels in (patho)physiological conditions are not clear yet. We believe that deciphering the events associated with the regression of brain microvessels induced by plastic changes in a healthy brain or chronic neurodegeneration and aging would open up new approaches to the effective restoration of cognitive deficits. In addition, it is of particular interest for the development of functionally competent brain tissue in vitro models, including those based on iPSCs-derived cerebral organoids, or functional BBB models suitable for testing new drug candidates.

## Figures and Tables

**Figure 1 ijms-23-12683-f001:**
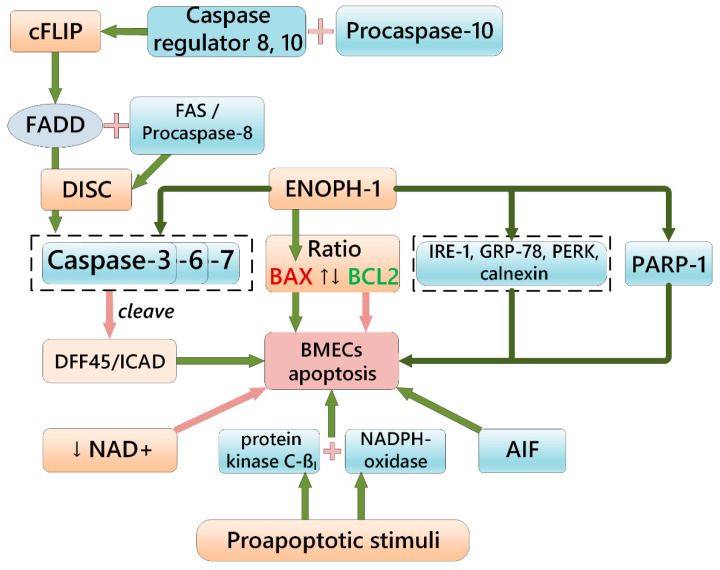
Scheme illustrating key events in the apoptosis induction and progression in BMECs.

**Figure 2 ijms-23-12683-f002:**
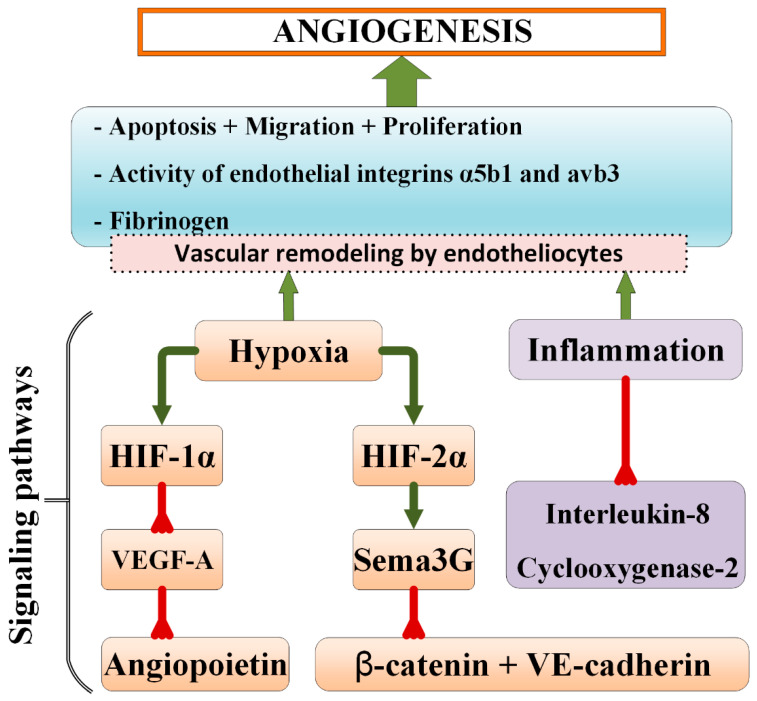
Scheme illustrating some pharmacological targets used for normalizing brain vascular remodeling.

## Data Availability

Not applicable.

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
