# Peer review of "Physiological and Pathological Remodeling of Cerebral Microvessels"

_ijms, 2022, doi:10.3390/ijms232012683_

Round 1

Reviewer 1 Report

Overall, this review is on an interesting topic and brings together several lines of evidence. The main concern with the manuscript is that the flow of logic and transitions make reading difficult. 

Minor

·   1      Several sentences do have adequate referencing, especially in the mechanism section

·  2       Manuscript has grammatical errors and the writing is unclear. The flow of logic is hard to hard. What is written is mainly bullet-form of various studies with poor transitions between sentences/ideas. A professional editing company should be used to revise the manuscript.

Author Response

Dear Reviewer,

On behalf of all co-authors, I would like to thank for your comments and suggestions for the

improvement of our manuscript. All suggestions and comments were regarded and studied.

All changes were highlighted in the manuscript.

Best regards,

sincerely yours,

Dr. Pavel Tegub

(on behalf of all authors)

Overall, this review is on an interesting topic and brings together several lines of evidence. The main concern with the manuscript is that the flow of logic and transitions make reading difficult.

- Corrections of the scientific language have been made to the manuscript.

Minor

  • 2 Manuscript has grammatical errors and the writing is unclear. The flow of logic is hard to hard. What is written is mainly bullet-form of various studies with poor transitions between sentences/ideas. A professional editing company should be used to revise the manuscript.

- Corrections of the scientific language have been made to the manuscript.

Reviewer 2 Report

Some minor revisions for the paper:

1) The term "neoangiogensis" is not commonly used and could generate confusion, it should be clearly defined early in the article

2) a few sentences have a challenging sentence structure, if possible an editorial read by a native english speaker could improve readability

3) typo on line 85, presumably "VIEF Receptors" should read "VEGF Receptors"

4) The section on VEGF signalling could be strengthened by adding a short discussion of the  key physiological / signalling differences of the 3 main types of VEGF Receptors

5) The discussion of string vessels is great, however a small description of the estimated lifetime of string vessels and the remaining uncertainties would help (William R Brown has written several excellent reviews on the topic)

6) On line 472 the Author's initial should precede their last name ie O. Bracko and M. Ali

Author Response

Dear Reviewer,

On behalf of all co-authors, I would like to thank for your comments and suggestions for the

improvement of our manuscript. All suggestions and comments were regarded and studied.

All changes were highlighted in the manuscript.

Best regards,

sincerely yours,

Dr. Pavel Tegub

(on behalf of all authors)

Some minor revisions for the paper:

1) The term "neoangiogensis" is not commonly used and could generate confusion, it should be clearly defined early in the article

- Corrections have been made.

2) a few sentences have a challenging sentence structure, if possible an editorial read by a native english speaker could improve readability

- Corrections of the scientific language have been made to the manuscript.

3) typo on line 85, presumably "VIEF Receptors" should read "VEGF Receptors"

- Corrections have been made.

4) The section on VEGF signalling could be strengthened by adding a short discussion of the key physiological / signalling differences of the 3 main types of VEGF Receptors

- Additions have been made to the manuscript.

5) The discussion of string vessels is great, however a small description of the estimated lifetime of string vessels and the remaining uncertainties would help (William R Brown has written several excellent reviews on the topic)

- Additions have been made to the manuscript.

6) On line 472 the Author's initial should precede their last name ie O. Bracko and M. Ali

- Corrections have been made.

Round 2

Reviewer 1 Report

The manuscript still has significant grammatical errors. Please consult professional help to get these fixed.